# EVALUATING THE RETRIEVAL ROBUSTNESS OF LARGE LANGUAGE MODELS

## ABSTRACT

Retrieval-augmented generation (RAG) generally enhances large language models' (LLMs) ability to solve knowledge-intensive tasks. But RAG could also lead to performance degradation due to imperfect retrieval and the model's limited ability to leverage retrieved content. In this work, we evaluate the robustness of LLMs in practical RAG setups (henceforth *retrieval robustness*). We focus on three research questions: (1) whether RAG is always better than non-RAG; (2) whether more retrieved documents always lead to better performance; (3) and whether document orders impact results. To facilitate this study, we establish a benchmark of 1500 open-domain questions, each with retrieved documents from Wikipedia. We introduce three robustness metrics, each corresponds to one research question. Our comprehensive experiments, involving 11 LLMs and 3 prompting strategies, reveal that all of these LLMs exhibit surprisingly high retrieval robustness; nonetheless, different degrees of imperfect robustness hinders them from fully utilizing the benefits of RAG.

## 1 INTRODUCTION

Large language models (LLMs) learn to acquire massive amounts of knowledge through large-scale pre-training, enabling them to answer knowledge-intensive questions (OpenAI et al., 2024; Anthropic, July. 2024; Meta, September 2024). However, relying exclusively on parametric knowledge can lead to inaccuracies when dealing with unseen or time-sensitive information, or when the model fails to precisely retrieve relevant knowledge from its own parameters. To alleviate these limitations, retrieval-augmented generation (RAG) is proposed, where external documents containing information relevant to the task are fetched from a datastore and provided to the model as context during inference (Chen et al., 2017; Lewis et al., 2020).

Despite its potential, RAG does not always guarantee performance improvements. The retriever might fail to retrieve relevant documents, and the LLMs might be distracted by irrelevant content, leading to performance drop (Mallen et al., 2023). As achieving a perfect retriever remains an elusive goal in practice, it is crucial for LLMs to behave robustly in the RAG setup to reduce the risks during actual deployment.

Previous work has shown that LLMs are particularly vulnerable when provided with noisy contexts that are synthetically constructed (Chen et al., 2024). Distracted by the specially designed misleading information, models tend to produce incorrect outputs (Wu et al., 2024b). Despite yielding valuable insights, synthetically constructed contexts are dissimilar to realistic retrieved contexts that are usually drawn from credible corpora like Wikipedia and trusted news outlets.

To bridge this gap, this work benchmarks LLMs' robustness under realistic RAG setups. We consider an LLM *retrieval robust* if (1) its RAG performance is equal to or better than its non-RAG performance; (2) adding more retrieved documents leads to equal or better performance; and (3) its RAG performance is invariant to the order of retrieved documents. Three metrics are defined correspondingly—no-degradation rate, retrieval size robustness, and retrieval order robustness.

We focus on open-domain question answering—a knowledge-intensive task where RAG is widely adopted. We curate a benchmark of 1,500 samples by randomly drawing 500 questions each from three datasets—Natural Questions (Kwiatkowski et al., 2019), Hotpot QA (Yang et al., 2018), ASQA (Stelmakh et al., 2022)—covering diverse domains and complexities.

To construct retrieved contexts, we leverage two retrievers, including a canonical sparse BM25 (Robertson & Zaragoza, 2009) retriever and a dense retriever based on a strong embedding model, BGE (Xiao et al., 2023). Both retrievers retrieve context from Wikipedia articles. For analyses of retrieval size and order robustness, RAG setups with multiple retrieval sizes (5 to 100 documents) and three ways of ordering them (original rank, reversed rank, random shuffle) are evaluated. Our experiments cover 11 LLMs from both open-source and proprietary families. Each LLM is evaluated via vanilla prompting and two more sophisticated prompting strategies: one augments the model's own knowledge, and the other filters relevant retrieval contexts.

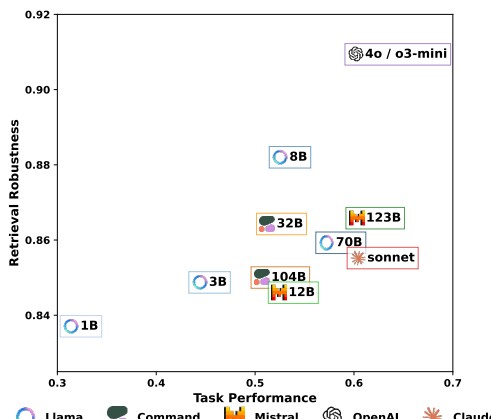

Figure 1: Comparison of retrieval robustness and QA task performance across various LLMs. The y-axis represents robustness (geometric mean of the three robustness metrics), while the x-axis represents task performance (average across all $k$, $o$, retrievers, and datasets). OpenAI GPT-4o and o3-mini have very close robustness and performance.

We find that LLMs are quite robust in general, achieving over 80% scores on the geometric mean of the three retrieval robustness metrics, as shown by Figure 1. This indicates that, *oftentimes*, (1) RAG is better than non-RAG; (2) more retrieved documents lead to better performance; and (3) order of the documents does not matter a lot. Nonetheless, the imperfect retrieval robustness reflects undesired behaviors, notably the performance trade-off among individual samples (i.e., hurting performance on some examples while gaining performance on others), which prevents the models from fully utilizing the benefits of RAG and destabilizes response quality when changing the retrieval size or order. Such unpredictable trade-off poses risks for realistic applications that demand consistent outcomes. Finally, we find that retrieval robustness can be enhanced by augmenting the answers generated with the model's own knowledge, though it also limits the potential task performance gain from RAG.

Our contributions are summarized as follows:

- We propose sample-level metrics to rigorously measure *retrieval robustness*—how robust LLMs handle queries in RAG setups.
- We compile a benchmark for evaluating retrieval robustness, following common RAG setups in practice. It comprises diverse open-domain QA tasks along with retrieved documents from Wikipedia obtained by widely-used and strong retrievers.
- We conduct a comprehensive empirical study of 11 modern LLMs with 3 different prompting strategies, revealing the generally good robustness of LLMs in more realistic settings and highlighting the consequences of their imperfect robustness.

## 2 RELATED WORKS

**Retrieval-Augmented Generation (RAG)** enhances parametric models by retrieving semantically relevant information from a knowledge base (Gao et al., 2023b; Wu et al., 2024a). Typically, it involves a retriever and a parametric language model. RAG can potentially help adapt pretrained models to up-to-date knowledge, ground models with long-tail information, and thus improve factuality and accuracy (Asai et al., 2024). The pioneering RAG framework, DrQA (Chen et al., 2017), was introduced to tackle knowledge-intensive open-domain question answering (QA) tasks, which is still the main evaluation target of recent works. RAG has also been used for non-knowledge-intensive tasks like language modeling, understanding, and reasoning (Borgeaud et al., 2022; Guo et al., 2023; Izacard et al., 2024). There are many different ways to implement RAG. Some works, e.g., knn-LM (Khandelwal et al., 2020), retrieve hidden states, while many other works retrieve text. To utilize the retrieved documents, some works modified the model architecture. e.g., FiD (Izacard & Grave, 2021) encoded each document separately and concatenated their hidden states in the decoder, while RETRO (Borgeaud et al., 2022) added a chunked cross-attention module into the

regular Transformer block. Another widely used method is to simply include the retrieved documents directly into the input. This can be done by putting them all together in one context (Ram et al., 2023; Lee et al., 2024) or by generating answers with each of them separately and ensembling the results (Guu et al., 2020; Lewis et al., 2020; Shi et al., 2024). Some works train the retriever and the language model jointly (Lewis et al., 2020; Borgeaud et al., 2022; Lin et al., 2024), while others fix the model and and only train the retriever (Ram et al., 2023; Shi et al., 2024). In this paper, we opt for the simplest setup: we use off-the-shelf retrievers and LLMs, and we use the retrieved documents by directly including them in a single context window. This approach has become increasingly practical with the long-context ability of modern LLMs (Lee et al., 2024).

**Retrieval Robustness.** Neural language models are shown to be easily distracted by adversarially inserted irrelevant content (Jia & Liang, 2017; Shi et al., 2023; Weston & Sukhbaatar, 2023). However, irrelevant context comes in naturally in any RAG setup due to the imperfect retriever. Chen et al. (2024) showed that the LLM-based RAG performance goes down when increasing the noise (i.e., documents that are relevant to the question but do not contain any information about the answer) rate. Wu et al. (2024b) conducted a deeper analysis and found that highly semantically related information is more likely to distract LLMs. Thakur et al. (2024) evaluated LLM RAG performance with a completely irrelevant set of documents and observed non-trivial hallucination rates. Yoran et al. (2024) introduced the concept of *retrieval robustness*, "retrieval-robust LLMs states that: (a) when relevant, the retrieved context should improve model performance; (b) when irrelevant, the retrieved context should not hurt model performance." However, all these works usually handcrafted controlled yet synthetic evaluation setups that mixing irrelevant context with relevant ones. Following the same spirit, we instead resort to a more realistic and practical setup where we simply pick the top-$K$ contexts returned by a retriever which a natural mixture of relevant and irrelevant content. And we extend the definition of *retrieval robustness* to the three conditions stated in the introduction. In addition, some recent works tried to make RAG robust to intentional knowledge corruption attacks, e.g., injecting malicious facts (Zou et al., 2024; Anonymous, 2024), which is not the type of robustness we would like to evaluate in this paper.

## 3 ROBUSTNESS METRICS

In this section, we present the three critical metrics for evaluating the retrieval robustness of an LLM system, illustrated in Figure 2. We define an LLM system as a backbone LLM, paired with a prompting strategy. Let $f(q, k, o)$ denote the performance of an LLM system, where $q$ is the task query, $k$ is the number of retrieved documents, and $o$ specifies the order of the retrieved documents. In this paper, $f(q, k, o)$ is the correctness of the model's response to $q$, assessed by an LLM judge by comparing with the reference answer (§4.1). When $k > 0$, $f(q, k, o)$ represents the performance of the LLM system in the RAG setup. For consistency, we use $f(q, 0)$ to denote the performance of the LLM system in the non-RAG setup, where model answers the query using its own knowledge. See Section 4.3 for the choices of $k$ and $o$ in our experiments.

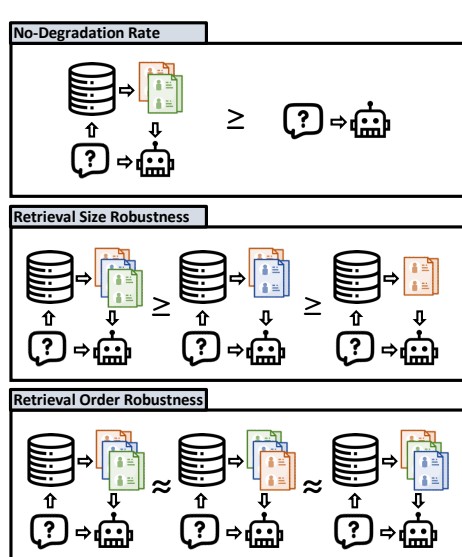

Figure 2: Our retrieval robustness metrics, targeting three research questions: (1) whether RAG is always better than non-RAG; (2) whether more retrieved documents always lead to better performance; (3) whether document orders lead to consistent results.

**No-Degradation Rate (NDR).** This metric measures how often the LLM system's performance with RAG $f(q, k, o)$ (for any $k > 0$ and $o$) is at least as good as the performance without

RAG $f(q, 0)$, which is calculated as:

$$\text{NDR} = \frac{1}{Z} \sum_{q \in Q} \sum_{k \in K} \sum_{o \in O} \mathbb{1}\big[f(q, k, o) \geq f(q, 0)\big] \tag{1}$$

where $K$ includes all choices of numbers of retrieved documents, $O$ represents all possible document orders used in the benchmark, and $Q$ is the set of all task samples. $Z = |Q| \cdot |K| \cdot |O|$ is the normalization factor for the aggregation. A high NDR implies that, for most queries, using retrieval does not degrade performance relative to the non-RAG baseline.

**Retrieval Size Robustness (RSR).** This metric examines how the system behaves as the number of retrieved documents increases. Specifically, for each task query $q$ and each value of $k$, we check whether the performance is maintained or improved, compared to all smaller values of $k$. RSR only considers $k > 0$, not involving the effect of NDR. Results for various $k$s are then aggregated across all task samples, formally defined as:

$$\text{RSR}_{(q, k_i, o)} = \mathbb{1}\big[\wedge_{j<i}[f(q, k_i, o) \geq f(q, k_j, o)]\big]$$
$$\text{RSR} = \frac{1}{Z} \sum_{q \in Q} \sum_{k_i \in K, i > 1} \sum_{o \in O} \text{RSR}_{(q, k_i, o)} \tag{2}$$

where $Z = |Q| \cdot (|K| - 1) \cdot |O|$. A high RSR indicates that performance rarely degrades when adding more retrieved documents.

**Retrieval Order Robustness (ROR).** ROR concerns the sensitivity of the system to the order of the same set of retrieved documents. For a task sample $q$ and $k > 0$, let $O$ denote selected choices of permutations of the $k$ documents. We can compute the standard deviation of the model performance over all permutations $o \in O$, which is represented as $\sigma_{o \in O}[f(q, k, o)]$. For performance metrics bounded between 0 and 1, the standard deviation is bounded between 0 and 0.5. Therefore, we scale it by a factor of 2 to ensure the robustness metric ranges between 0 and 1. We compute the ROR score as:

$$\text{ROR} = \frac{1}{Z} \sum_{q \in Q} \sum_{k \in K} \big(1 - 2\sigma_{o \in O}\big[f(q, k, o)\big]\big) \tag{3}$$

where $Z = |Q| \cdot |K|$. A higher ROR means that different permutations of the same set of documents produce more consistent performance.

The three metrics capture complementary aspects of retrieval robustness, reflecting different desired behaviors of LLM systems with RAG in real world applications. NDR provides a safety guarantee that retrieval will not harm results; RSR is critical for scenarios where retrieval size can be scaled up for enhanced performance; and ROR is important for situations where document ranking is imperfect. Note that, for simplicity, we omit the marginalization over two different retrievers (see Section 4.3) from the equations of all three metrics.

## 4 BENCHMARK SETUPS

We conduct experiments to benchmark retrieval robustness of LLM systems. Though RAG can be applied for various tasks, we focus on the task where RAG is commonly adopted—answering knowledge-intensive open-domain questions.

### 4.1 DATA AND EVALUATION METRICS

**Open-domain QA Tasks.** We sample from three QA datasets. Natural Questions (Kwiatkowski et al., 2019) contains samples derived from Google Search queries, covering a broad range of questions real-world users ask online; Hotpot QA (Yang et al., 2018) is a multi-hop QA dataset that requires chaining multiple passages to answer questions; ASQA (Stelmakh et al., 2022) targets extraction of key information from multiple sources. We randomly sample 500 examples from each of the datasets, totaling 1500 samples.

**Evaluation Metrics.** Previous work usually used string match metrics for answers evaluation (Mallen et al., 2023; Gao et al., 2023a). However, it is rigid and can not evaluate model performance very well. Therefore, we prompt (see the prompts we used in Appendix D) Llama-3.3-70B-Instruct to evaluate whether the generated responses align with the gold answers.[1]

**Retrieval Corpus.** We use Wikipedia as the corpus to retrieve documents from. We processed the wikidump from June 2024, which contains 6 million articles. We split each article into chunks by double newlines, resulting in 20 million chunks. Each chunk is treated as an independent "document" for retrieval.

## 4.2 LLM Systems

**Backbone LLMs.** 11 LLMs from three open-source families and two proprietary families are tested, including Llama-3 Instruct (3.1-8B, 3.1-70B, 3.2-1B, 3.2-3B) (Meta, July 2024;S), Mistral Instruct (Nemo, Large) (Mistral.ai, July 2024;F), Command (R, R plus) (Cohere, Aug. 2024), OpenAI GPT-4o (OpenAI et al., 2024), o3-mini (OpenAI, 2025), and Claude-3.5-sonnet (Anthropic, July. 2024).

**Prompting Strategies.** Besides the vanilla prompting strategy that concatenates all retrieved documents in the prompt, we explore two alternative strategies that might help model incorporate information in the retrieved documents more robustly. Both strategies involve two steps. (1) **OwnKnow** obtains a draft answer based on models' own knowledge by prompting without retrieval in the first step, and then inserts this draft answer into the prompt for the RAG setup. (2) **S2A**, inspired by System 2 Attention (Weston & Sukhbaatar, 2023), first tries to identify the relevant retrieved documents in the first step, and then only uses the identified documents in the RAG setup. This decouples relevance estimation from answer extraction, allowing the answer extraction step to focus on the most pertinent information.

## 4.3 RAG Parameters

**Retrievers.** Our retrieval system is built on top of Solr 9[2]. We use two retrievers: one is the canonical sparse retriever based on BM25 (Robertson & Zaragoza, 2009), and the other is cosine similarity based dense retriever where we embedded each document by bge-large-en-v1.5[3] (Xiao et al., 2023). For any robustness metric defined in Section 3, we get the results for both retrievers and take the average.

**Sizes.** We experiment with retrieval sizes of 5, 10, 25, 50, 75, and 100 documents. The retrieval size is capped at 100 documents as most models have reached their maximum context lengths. When the retrieved documents exceed the maximum context length of a model, we iteratively drop the lowest ranked document.

**Orders.** For each of these sizes, we apply three ordering strategies based on the retriever's ranking of the documents: the **original** order (returned by the retriever), the **reversed** order

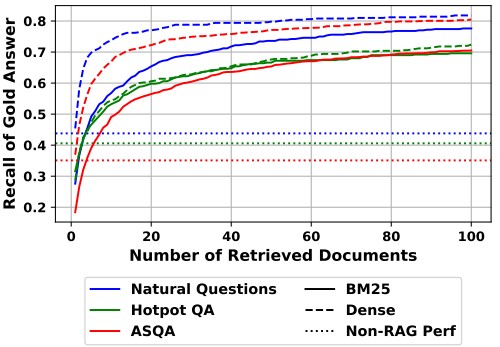

Figure 3: Performance of the retrievers, measured by the recall of gold answers within the concatenated retrieved documents. The gold answer is considered covered if any of its alternative forms exactly match a substring in the concatenated retrieved documents.

---

[1]We also tried GPT-4o as the judge initially. However due to cost constraints for large-scale evaluation, we opt for Llama-3.3-70B-Instruct. And on a subset of 2,000 samples, we find these two models agree at 93% of time.

[2]https://solr.apache.org/docs/9_0_0/index.html

[3]http://huggingface.co/BAAI/bge-large-en-v1.5

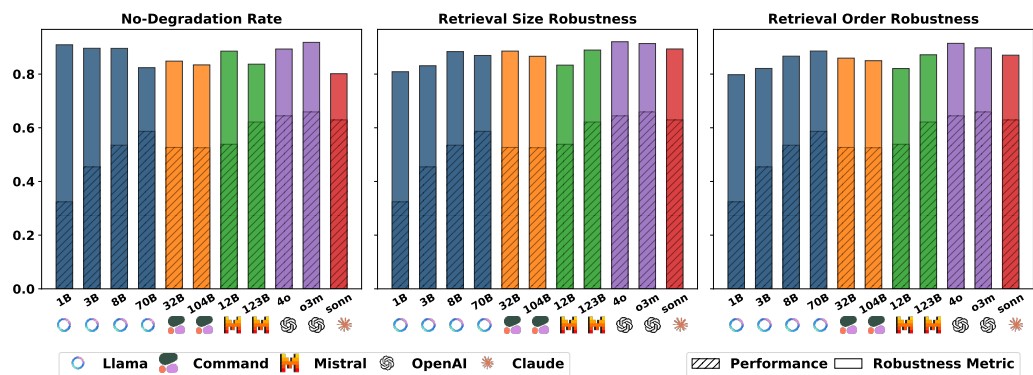

Figure 4: The three retrieval robustness metrics and task performance of experimented LLMs using vanilla prompting. o3m: o3-mini; sonn: sonnet. The mean of task performance achieved with different retrieval sizes and orders are is shown for each model. Model families are indicated by icons, while the variants are indicated by model sizes (except for GPT-4o and Claude-3.5-sonnet). As Llama variants of different sizes are released in different versions, Llama-3.1 and Llama-3.2 are both included. Models generally have good retrieval robustness. While larger model sizes lead to improved task performance, there exists no consistent trend across the retrieval robustness metrics.

(reversing the original order), and a randomly **shuffled** order. We test the reversed order because sometimes we want to put the most relevant document to the end of the prompt (the closest to the answer). We include a random order to simulate any potential reranking logic on top of the retriever.

**Retrieval Quality.** As our retrieval robustness benchmark relies on the retrievers, we examine the retrieval quality by checking the recall of gold answers within the retrieved documents. We follow prior work and determine if the concatenated retrieved documents contain the gold answer if its substring is an exact match of any form of the gold answer (substring exact match) (Mallen et al., 2023). For reference, we also report the best model performance without RAG (Non-RAG Perf) to highlight the potential improvement that can be obtained with RAG. As shown in Figure 3, both retrievers provide sufficiently high-quality retrieval, ensuring that the findings of our experiments are based on valid setups.

## 5 RESULTS

### 5.1 OVERALL ROBUSTNESS

We report the three retrieval robustness metrics for LLM systems using vanilla prompting in Figure 4. Besides robustness, task performance is shown in the same figure with bars with a different hatch style. Retrieval robustness is calculated following the definitions in Section 3, while task performance is the average score across all $k$, $o$, retrievers, and datasets. **All models achieve higher than 80% retrieval robustness across all metrics, with GPT-4o and o3-mini surpassing 90%.** Compared to prior studies that highlight the weak robustness of RAG systems under synthetic setups, such as using artificially created documents (Wu et al., 2024b), we show that LLMs demonstrate surprisingly good retrieval robustness in more realistic settings. This high retrieval robustness means we can safely apply RAG without overly stressing about whether RAG is better than non-RAG and about the decisions on retrieval size and order, which can potential simplify RAG systems. Nevertheless, the remaining 10% may pose challenges for real-world deployment, particular for high-stake domains where comprehensive reliability is required.

### 5.2 RELATION BETWEEN ROBUSTNESS AND PERFORMANCE

Although retrieval robustness metrics are derived from the sample-level task performance, retrieval robustness does not always correlate with task performance. As shown in Figure 1 and Figure 4, task performance usually gets better when models get larger. In contrast, we note that, **larger LLMs can**

**have lower retrieval robustness than smaller LLMs**. For example, in Figure 1, Llama-3-8B has higher robustness than 70B. If we "zoom in" to each of the three robustness metrics (Figure 4), we can see that this inverse scaling trend mainly comes from No-Degradation Rate (NDR). This is because larger models usually have richer parametric knowledge and answers more questions correctly without retrieval, which means RAG will have a higher baseline to beat and thus RAG is more likely to get worse than non-RAG. Therefore, in practice, when we apply RAG to knowledge-rich LLMs (usually models of larger sizes), we need to be cautious about whether it will lead to performance degradation from non-RAG.

Here, we use one example to show how **low robustness reduces RAG efficacy**. In Figure 5, solid lines illustrate the actual performance of Mistral-Large and o3-mini at different number of retrieved documents. Dashed lines show their hypothetical performance under an oracle setup. This oracle setup assumes *perfect NDR*, meaning the models consistently generate responses at least as good as those produced without retrieval. As the solid lines show, although Mistral-Large surpasses o3-mini without retrieval (0 retrieved documents), it yields worse performance than o3-mini and even its own non-RAG baseline when RAG is applied. Conversely, if Mistral-Large has perfect NDR, it would outperform o3-mini in the RAG setup. The gap between the actual and oracle setups demonstrate that Mistral-Large fails to preserve its non-RAG performance for approximately 14% of the dataset samples, due to the insufficient retrieval robustness. Overall, retrieval robustness metrics **complement** standard task performance metrics and provide a new perspective of how well LLMs perform in RAG settings.

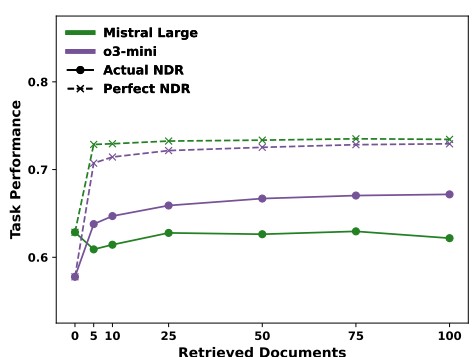

Figure 5: Task performance of models using vanilla prompting under setups with actual no-degradation rate (NDR) and perfect NDR. Enhancing retrieval robustness could lead to a 12% absolute performance gain for both models.

### 5.3 EFFECT OF RETRIEVAL SIZE

For most of the models, the overall **task performance is generally increasing as more retrieved documents are added** (see Figure 13, 14, 15, and 16 in Appendix). This again demonstrates that in practice we do not have to overly concern about picking the optimal retrieval size. If budget allows, we can simply keep adding more documents till it reaches the max input length limit.

Nevertheless, this does not indicate perfect retrieval size robustness, as **models keep trading off performance across individual samples**, i.e., hurting performance on some examples while gaining performance on others. Similar to the perfect NDR setup, we investigate an oracle setup with perfect RSR—choosing the best answer among those generated at current

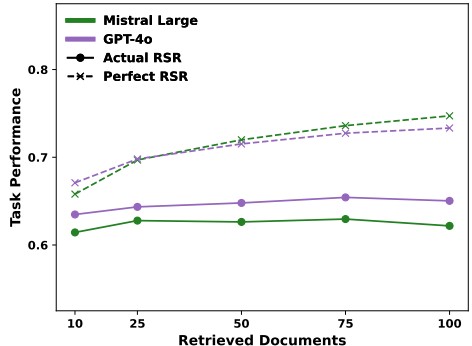

Figure 6: Task performance of models using vanilla prompting under setups with actual RSR and perfect RSR.

and all preceding values of $k$s as the final answer (Figure 6). Note that only answers produced by RAG (i.e., $k > 0$) are considered in the perfect RSR setup to eliminate the effect of NDR. Although, in the normal setup (actual RSR), task performance is increasing from $k = 10$ to $k = 75$, the gain is much more significant in the hypothetical perfect RSR situation, enlarging the gap between the two setups. This implies that models are constantly sacrificing some samples while enhancing others with larger retrieval sizes. We think that the increasing number of retrieved documents chal-

lenges models' ability to identify helpful documents and handle longer inputs, and thus leads to the imperfect robustness on retrieval size.

## 5.4 EFFECT OF RETRIEVAL ORDER

We break down retrieval robustness and task performance by the order of the retrieved documents (Figure 7). Overall, **LLMs demonstrate good retrieval order robustness – the performance achieved with different orders of the retrieved documents is similar**. This means, in practice, we do not have to overly concern about the order of documents. While GPT-4o and o3-mini demonstrate the strongest retrieval robustness and performance with normally ordered documents, all other models prefer the reversed order. This suggests that **placing higher-ranked retrieved documents closer to the question** generally optimizes RAG performance (see the prompt `rag_qa.j2` in Appendix D).

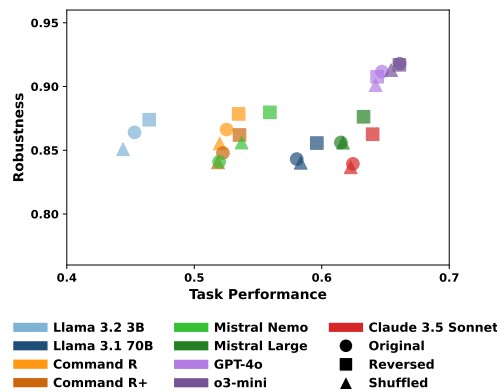

Figure 7: Geometric mean of no-degradation rate and retrieval size robustness, grouped by the order of retrieved documents.

Despite this high robustness, we underscore that **performance variance across orders persists at the sample level**. We establish an oracle setup for retrieval order robustness that selects the best response among responses generated with retrieved contexts ordered differently (*perfect ROR*), as shown in Figure 8. Picking the best response for each example across different orders exhibits a large performance gain from each individual document order. This indicates that each example has a different *best* order, highlighting the need for continuing efforts to improve order robustness.

## 5.5 EFFECTS OF PROMPTING STRATEGIES

Using prompting strategies to decompose response generation has demonstrated effectiveness in handling complex tasks. Figure 9 shows that only the **OwnKnow** strategy that incorporates answers generated in the non-RAG setup can consistently enhance retrieval robustness. We believe outputs given by the non-RAG setup serve as drafts and anchors, leading to reduced variance. It is also possible that **OwnKnow** benefits from its similarity to self-refinement that was shown to be an effective prompting technique (Yang et al., 2022; Madaan et al.,

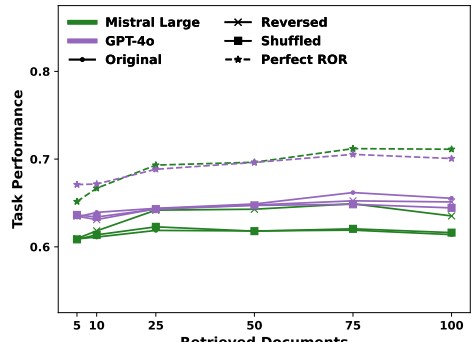

Figure 8: Task performance of models using vanilla prompting under setups with actual ROR for each order and perfect ROR.

2023). Although selecting task-relevant context benefits robustness when synthetic noisy passages are injected into the input as shown by Weston & Sukhbaatar (2023), a similar **S2A** prompting strategy fails to enhance retrieval robustness in our evaluations. We conjecture that, compared to synthetic noisy contexts, realistic retrievers provide models with harder negative contexts that are more challenging for the model to identify.

As we look into the maximum task performance across retrieval sizes rather than the mean task performance, we observe that using **OwnKnow** might limit the maximum performance models can possibly achieve, suggesting that the higher retrieval robustness of **OwnKnow** comes at a cost of RAG effectiveness.

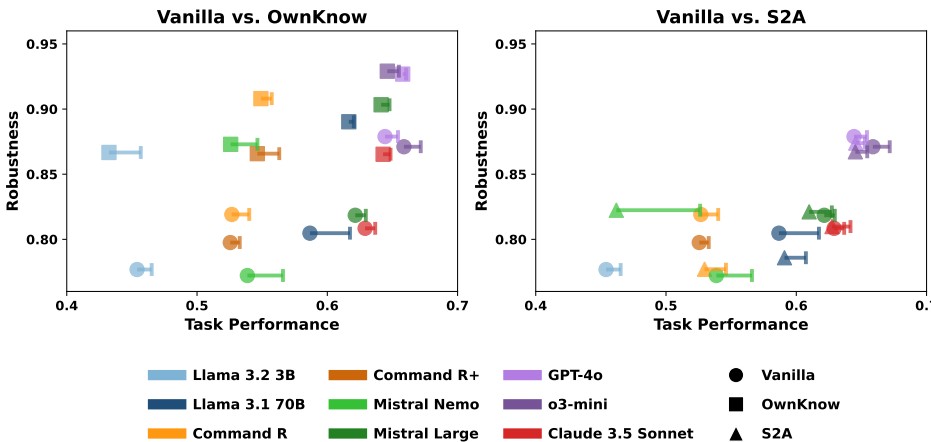

Figure 9: Geometry mean of the three retrieval robustness metrics and task performance of LLMs paired with different prompting strategies. The mean of task performance achieved with different retrieval sizes and orders are shown for each model. Models are differentiated with colors and prompting strategies are indicated by marker styles. The bar on the right of each marker indicates the maximum performance across retrieval sizes.

## 6 CONCLUSIONS

We introduce retrieval robustness metrics—no-degradation rate, retrieval size robustness, and retrieval order robustness—to quantify how reliably LLMs handle queries via RAG. A realistic benchmark of 1,500 questions is compiled, spanning three open-domain QA datasets, with augmented documents retrieved from Wikipedia using both sparse and dense retrievers. Our experiments with 10 LLMs from 5 families reveal that while models exceed 80% on those metrics, further improving retrieval robustness is a challenge beyond model scaling. Imperfect robustness result in sample-level trade-offs, often hurting the performance of some samples for the improvement on others, which forfeits RAG's potential gains. While incorporating outputs generated with the model's own knowledge can enhance retrieval robustness, it also limits the best performance that can be achieved by RAG. We hope our benchmark inspires further research on robust RAG systems.

## REPRODUCIBILITY STATEMENT

Questions in our benchmark come from Natural Questions Kwiatkowski et al. (2019) (Huggingface[4]), HotpotQA Yang et al. (2018) (Huggingface[5], and ASQA Stelmakh et al. (2022) (Subset of ALCE[6]). Upon acceptance, we will release scripts to reproduce our benchmark, including sampling questions from the three QA datasets, processing Wikipedia dump, obtaining retrieved documents based on the processed dump, and calculating metrics based on model outputs. We include the prompt templates we use in our experiment in Appendix D.

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

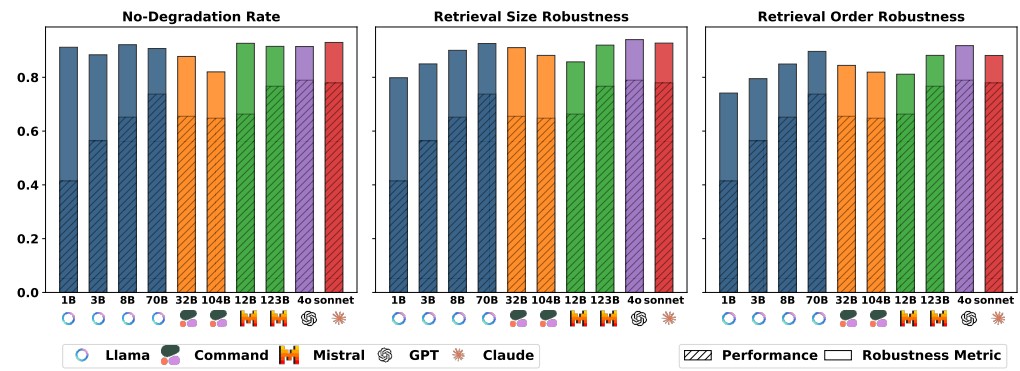

Figure 10: The three retrieval robustness metrics and task performance of experimented LLMs using vanilla prompting on Natural Questions.

Shitao Xiao, Zheng Liu, Peitian Zhang, and Niklas Muennighoff. C-pack: Packaged resources to advance general chinese embedding, 2023.

Kevin Yang, Yuandong Tian, Nanyun Peng, and Dan Klein. Re3: Generating longer stories with recursive reprompting and revision. In Yoav Goldberg, Zornitsa Kozareva, and Yue Zhang (eds.), *Proceedings of the 2022 Conference on Empirical Methods in Natural Language Processing*, pp. 4393–4479, Abu Dhabi, United Arab Emirates, December 2022. Association for Computational Linguistics. doi: 10.18653/v1/2022.emnlp-main.296. URL https://aclanthology.org/2022.emnlp-main.296/.

Zhilin Yang, Peng Qi, Saizheng Zhang, Yoshua Bengio, William Cohen, Ruslan Salakhutdinov, and Christopher D. Manning. HotpotQA: A dataset for diverse, explainable multi-hop question answering. In Ellen Riloff, David Chiang, Julia Hockenmaier, and Jun'ichi Tsujii (eds.), *Proceedings of the 2018 Conference on Empirical Methods in Natural Language Processing*, pp. 2369–2380, Brussels, Belgium, October-November 2018. Association for Computational Linguistics. doi: 10.18653/v1/D18-1259. URL https://aclanthology.org/D18-1259/.

Ori Yoran, Tomer Wolfson, Ori Ram, and Jonathan Berant. Making retrieval-augmented language models robust to irrelevant context. In *The Twelfth International Conference on Learning Representations*, 2024. URL https://openreview.net/forum?id=ZS4m74kZpH.

Wei Zou, Runpeng Geng, Binghui Wang, and Jinyuan Jia. Poisonedrag: Knowledge corruption attacks to retrieval-augmented generation of large language models, 2024. URL https://arxiv.org/abs/2402.07867.

## A ADDITIONAL RESULTS

### A.1 DATASET BREAKDOWN OF RETRIEVAL ROBUSTNESS

We show the retrieval robustness metrics and average RAG performance in Figure 10, 11, and 12. Across all individual datasets, there is still no consistent improvement in retrieval robustness with increased model sizes.

### A.2 DATASET BREAKDOWN OF RAG PERFORMANCE ACROSS $k$S

We show open-domain QA performance at different numbers of retrieved documents in Figure 13, with dataset breakdown in Figure 14, 15, and 16. Performance with each retriever and document order can be found in Figure 17, 18, and 19.

Compared to non-RAG, open-source LLMs with RAG can always boost performance, with the exception of Command R+ on Natural Questions. We also observe a performance drop on Hotpot QA with the dense retriever when using Llama-3.1-70B.

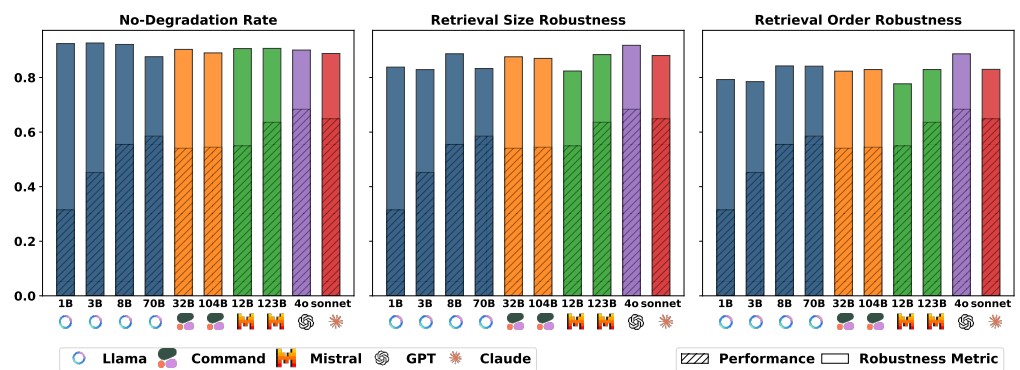

Figure 11: The three retrieval robustness metrics and task performance of experimented LLMs using vanilla prompting on Hotpot QA.

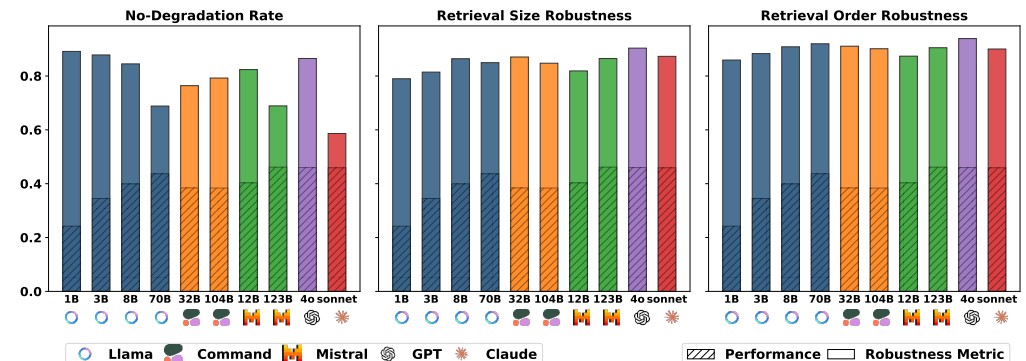

Figure 12: The three retrieval robustness metrics and task performance of experimented LLMs using vanilla prompting on ASQA.

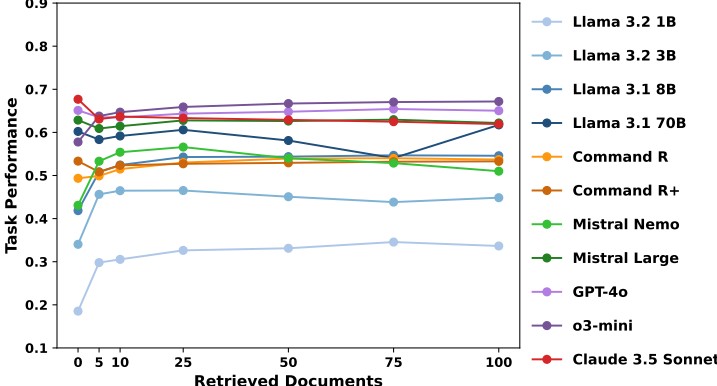

Figure 13: Performance averaged across datasets, retrievers, and document orders.

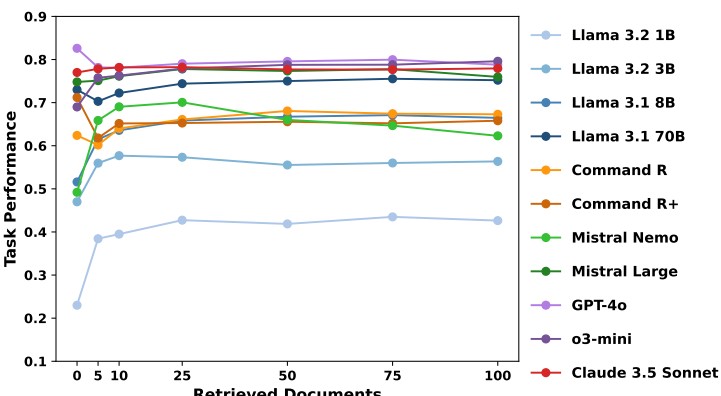

Figure 14: Performance on Natural Questions, averaged across retrievers and document orders.

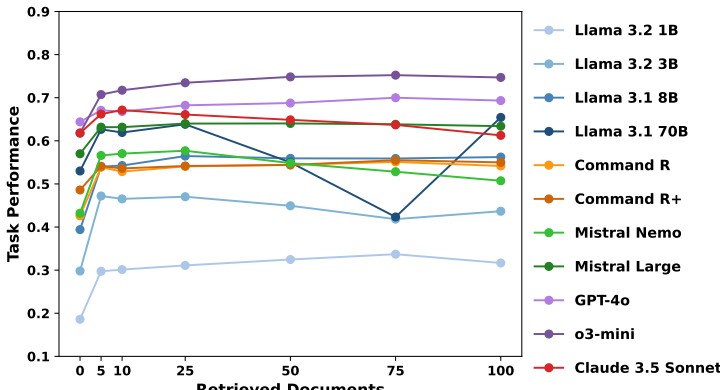

Figure 15: Performance on Hotpot QA, averaged across retrievers and document orders.

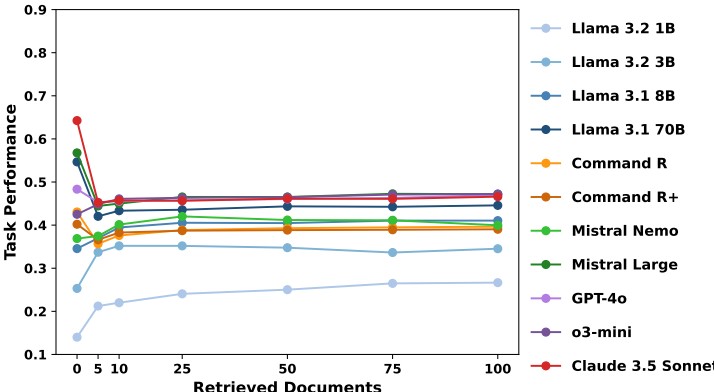

Figure 16: Performance on ASQA, averaged across retrievers and document orders.

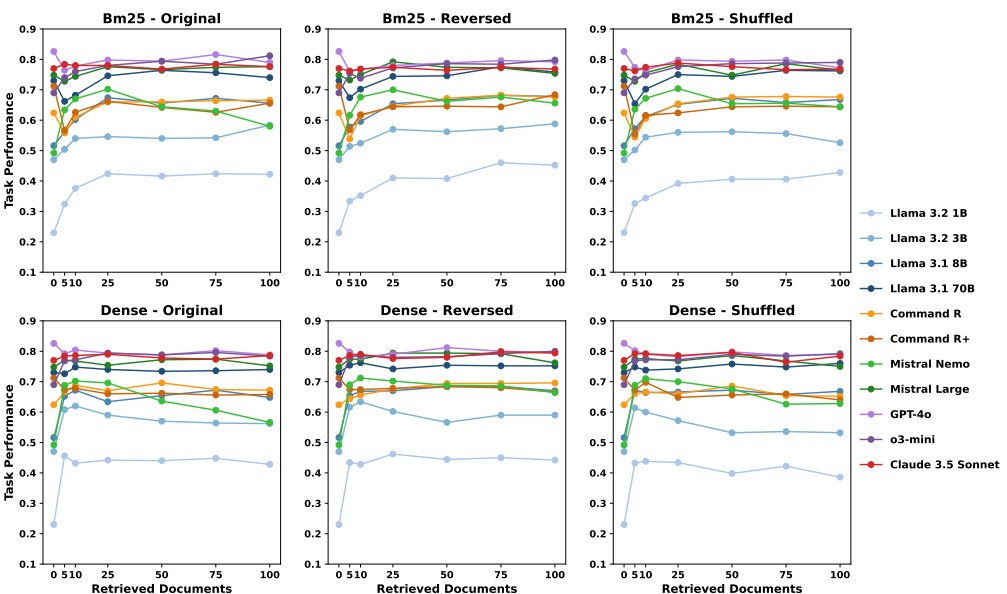

Figure 17: Performance on Natural Questions with different retrievers and document orders.

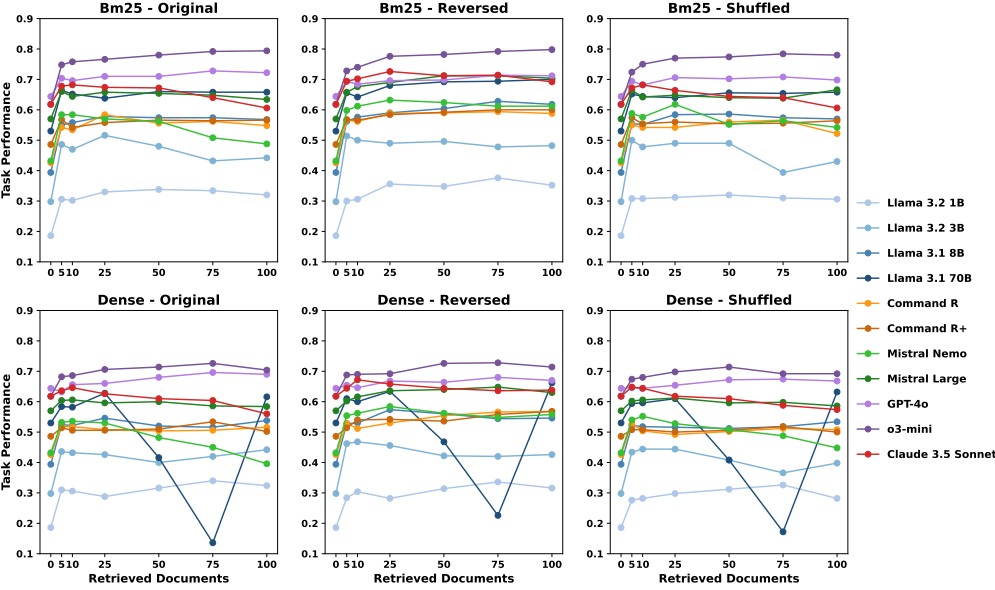

Figure 18: Performance on Hotpot QA with different retrievers and document orders.

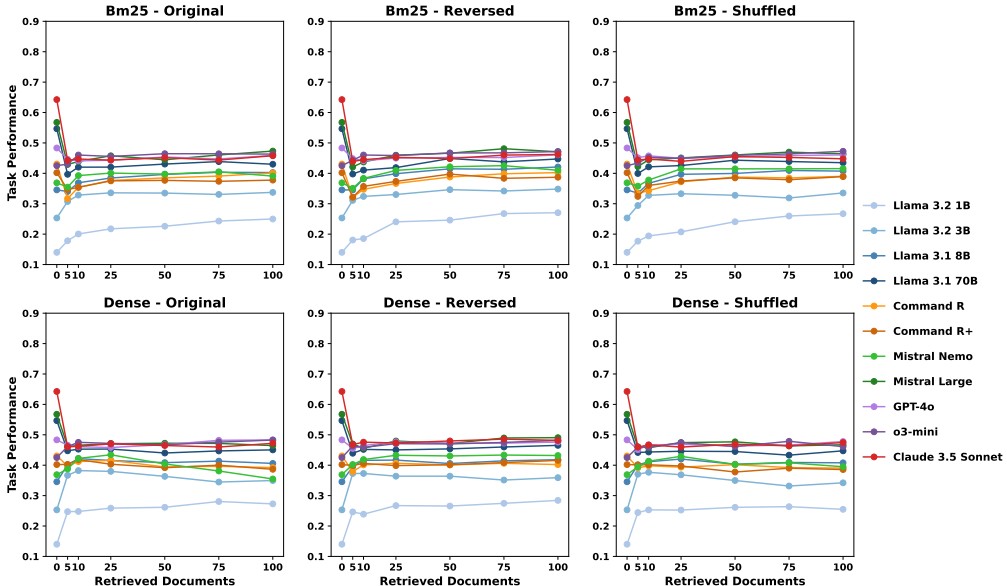

Figure 19: Performance on ASQA with different retrievers and document orders.

# B  INFERENCE SETUP

**Inference Parameters.**   Due to the computational cost and running time, we use greedy decoding and perform inference with each model under each setup once. During inference, models are allowed to generate at most 100 tokens, though they never exceed the limit.

**Inference Infrastructure.**   We use vLLM for more efficient inference (Kwon et al., 2023) and our experiments are conducted on compute nodes with 8 H100 GPUs.

# C  THE USE OF LLMS

We use LLMs (Gemini and ChatGPT) to polish writing, including correcting grammar errors and make language more concise.

# D  PROMPT TEMPLATES

The prompt templates (in jinja2 format) used in our experiments can be found at the end of Appendix.

`NON_RAG_QA.J2`

```
1 Answer the following question in a concise manner without explanation.
    Indicate your answer with "Answer:" and only include the answer words
    or phrases. For example: "Question: What city is Kowloon a part of?
    Answer: Hong Kong."
2
3 {{ question }}
```

`RAG_QA.J2`

```
1 Based on your own knowledge and retrieved contexts, answer the question
    in a concise manner without any explanation. Indicate your answer
    with "Answer:". For example: "Question: What city is Kowloon a part
    of? Answer: Hong Kong." If the answer is not specified or mentioned
    in the retrieved context, you must ignore the context and provide an
    answer by yourself. You must not refrain from answering the question.
2
3 Retrieved contexts:
4 {% for c in sources %}Context {{loop.index}}
5 {{c}}
6 {% endfor %}
7 {{ question }}
```

`OWNKNOW.J2`

```
1 Previously, you answer the question with your own knowledge. Now, based
    on your own knowledge and additional retrieved contexts, answer the
    question in a concise manner without any explanation. Indicate your
    answer with "Answer:". For example: "Question: What city is Kowloon a
    part of? Previous Answer: previous answer. Answer: Hong Kong." If
    the answer is not specified or mentioned in the retrieved context,
    you must ignore the context and provide an answer by yourself. You
    must not refrain from answering the question.
2
3 Retrieved contexts:
4 {% for c in sources %}Context {{loop.index}}
5 {{c}}
6 {% endfor %}
7 {{ question }} Previous Answer: {{ non_rag_output }}.
```

`S2A.J2`

```
1 Identify the retrieved context(s) that would be good context for
    providing an unbiased answer to the question. Indicate your selected
    context(s) "Selected Contexts:". For example: "Question: What city is
    Kowloon a part of? Selected Conetxts: Context 2, Context 5." If
    there is no retrieved context, reply with "Selected Conetxts: None".
2
3 Retrieved contexts:
4 {% for c in sources %}Context {{loop.index}}
5 {{c}}
6 {% endfor %}
7 {{ question }}
```

`ANSWER_EVALUATION_NQ_HOTPOT.J2`

```
1 You will be given a question, a list of gold answers to this question,
    and a predicted answer. Any one answer or multiple answers from the
    gold answer list can correctly answer the question. Your task is to
    judge whether the predicted answer can answer the question correctly.
2 Note that predicted answer does not have to exactly match one or multiple
    gold answers. It can answer the question correctly as long as its
    meaning entails one or multiple gold answers and there is no any
    additional incorrect information.
```

```
3
4 Question:
5 {{ question }}
6
7 Gold Answers:
8 {{ gold_answer }}
9
10 Predicted Answer:
11 {{ pred_answer }}
12
13 Is the predicted answer a correct answer to the question?
14
15 IMPORTANT: Please strictly follow the following format in your response:
16 [Start answer]
17 <Your answer. Choose from: Yes, No>
18 [End answer]
```

ANSWER_EVALUATION_ASQA.J2

```
1 You will be given a question, gold answers to this question, and a
       predicted answer. Gold answers are composed of multiple groups. Your
       task is to judge whether the predicted answer cover each group of the
        gold answers. Within one gold answer group, there can be multiple
       alternative answers. As long as one of the alternative answers is
       covered, the group is covered. Note that "cover" means "entail", in
       other words, you need to judge the predicted answer entails any
       answer within each group.
2
3 Question:
4 {{ question }}
5
6 Gold Answers:
7 {% for group in short_answer %}Group {{loop.index}}: {{ group }}
8 {% endfor %}
9 Predicted Answer:
10 {{ pred_answer }}
11
12 Does the predicted answer cover each group of the gold answers?
13
14 IMPORTANT: Please strictly follow the following format in your response:
15 [Start answer]
16 {% for group in short_answer %}Group {{loop.index}}: <Your answer. Choose
       from: Yes, No>
17 {% endfor %}[End answer]
```

