# OpenReview forum: "Evaluating the Retrieval Robustness of Large Language Models"
_ICLR.cc/2026/Conference — Submitted to ICLR 2026_

### Official Review · Reviewer_VDE5 · 2025-10-26

**Soundness:** 1
**Presentation:** 2
**Contribution:** 1
**Rating:** 2
**Confidence:** 3

**Summary:**

This paper introduces an evaluation framework for the "retrieval robustness" of large language models (LLMs) used in Retrieval-Augmented Generation (RAG) systems. It investigates if RAG always improves performance and how results are affected by the number and order of retrieved documents. To do this, it contributes a new benchmark of 1500 open-domain questions and three novel robustness metrics: no-degradation rate, retrieval size robustness, and retrieval order robustness. Comprehensive experiments across 11 LLMs reveal that while models are surprisingly robust (often >80% on these metrics), their imperfect robustness creates sample-level trade-offs, forfeiting RAG's potential gains. The study concludes that improving this robustness is a key challenge that cannot be solved by model scaling alone.

**Strengths:**

This paper has the following strengths:

- It formalizes "retrieval robustness" through a novel set of sample-level metrics. By proposing concrete measures for no-degradation, retrieval size, and document order robustness, it provides the community with a rigorous and standardized methodology to empirically quantify how LLMs handle the inevitable imperfections of real-world RAG pipelines.

- It creates a practical benchmark specifically designed for evaluating RAG robustness. This benchmark is a valuable resource, as it mirrors common RAG setups (using diverse open-domain QA and strong, widely-used retrievers), enabling standardized testing, replication, and fair comparison of different models and retrieval strategies.

- It conducts a comprehensive empirical study covering 11 modern LLMs and 3 prompting strategies. This broad analysis provides an insightful snapshot of the current landscape, establishing that while LLMs are surprisingly robust, their "imperfect robustness" remains a critical bottleneck that limits RAG's potential and presents a key challenge for future research.

**Weaknesses:**

This paper has the following weaknesses:

- The paper fails to discuss or compare against "Astute RAG" [1], a highly relevant study that also explores imperfect retrieval robustness and leverages the model's intrinsic knowledge. This omission makes it difficult to assess the paper's novel contributions.

- The rationale for curating a new, small-scale benchmark (1500 samples) from existing datasets is unclear, as is the advantage over using the original, larger datasets. The limited sample size raises concerns about the generalizability and statistical robustness of the empirical findings.

- The reliability of the results is questionable, as they depend on Llama3.3-70B-Instruct for correctness judgments. Using a model that is not state-of-the-art for evaluation may introduce significant noise and inaccuracies into the core metrics.

[1] Wang, Fei, et al. "Astute rag: Overcoming imperfect retrieval augmentation and knowledge conflicts for large language models." arXiv preprint arXiv:2410.07176 (2024).

**Questions:**

- Regarding related work, could the authors please discuss "Astute RAG"? That paper also seems to address imperfect retrieval and the use of a model's internal knowledge, so a direct comparison would be necessary to clarify this paper's specific novel contributions.

- Could the authors expand on the rationale for creating a new 1500-sample benchmark? It's unclear why this was preferable to using the original, larger datasets, and there are concerns that this limited scale may not be sufficient to support statistically robust or generalizable conclusions.

- The core metrics depend on Llama3.3-70B-Instruct for correctness judgments. Given that this model is not state-of-the-art and may introduce evaluation errors, what steps were taken to validate its reliability as a judge? Have the authors considered using a more advanced LLM or human evaluation to confirm the accuracy of the computed metrics?

---

> ### Author Response · Authors · 2025-12-04
>
> > Re Weakness 1 and Question 1: The paper fails to discuss or compare against "Astute RAG" [1], a highly relevant study that also explores imperfect retrieval robustness and leverages the model's intrinsic knowledge. This omission makes it difficult to assess the paper's novel contributions.
>
> We would like to clarify that the Astute RAG paper is a modeling paper, while our paper is an evaluation paper. Our contribution is completely orthogonal to the Astute RAG paper: we propose a set of metrics to measure the robustness of RAG systems when retrieving various numbers of documents in different orders, while their paper proposes a new RAG pipeline.
>
> > Re Weakness 2 and Question 2: The rationale for curating a new, small-scale benchmark (1500 samples) from existing datasets is unclear, as is the advantage over using the original, larger datasets. The limited sample size raises concerns about the generalizability and statistical robustness of the empirical findings.
>
> The scale of our experiments is very large due to the large number of setups: we have 2 retrievers, 6 retrieval sizes, 3 retrieval orders, and 3 prompting strategies, resulting in 108 setups per model. At this scale, experimenting with the full datasets is infeasible, and thus we choose to subsample from 3 different datasets. Many existing papers, including the AstuteRAG paper you are referencing in Weakness 1 also use subsampled datasets.
>
> > Re Weakness 3 and Question 3: The reliability of the results is questionable, as they depend on Llama3.3-70B-Instruct for correctness judgments. Using a model that is not state-of-the-art for evaluation may introduce significant noise and inaccuracies into the core metrics.
>
> Please see our general response, where we should that the judge model and the judging results are very robustness and would not affect the trend where our results and conclusions are based.

---

### Official Review · Reviewer_gXTC · 2025-10-31

**Soundness:** 3
**Presentation:** 3
**Contribution:** 3
**Rating:** 4
**Confidence:** 3

**Summary:**

This paper aims to evaluate the retrieval robustness of LLMs in "practical RAG setups". The authors construct a benchmark based on the Wikipedia corpus and real retrievers (BM25, BGE) and propose three new robustness metrics (NDR, RSR, ROR). The paper presents a comprehensive experimental analysis of 11 mainstream LLMs across different retrieval sizes, orders, and prompting strategies.
Overall, this work proposes a set of valuable evaluation metrics and conducts a thorough and insightful empirical study around a specific clean corpus setting.

**Strengths:**

1. The most significant strength of this paper is its exhaustive experimental design. The authors systematically investigate the robustness of 11 mainstream LLMs under a natural retrieval setting using the Wikipedia corpus. The study covers a wide range of variables and the experiments are thorough. Compared to previous benchmark work that focused on "artificially" constructing various types of adversarial noise, this paper's setup (strong LLM + natural top-K retrieval) helps us, in the current stage of LLM development, to re-evaluate the robustness of SOTA LLMs in retrieval-based QA, especially in "knowledge conflict" scenarios (i.e., where their own parametric knowledge conflicts with imperfect retrieved knowledge).

3. The authors devise three reasonable and meaningful robustness metrics (NDR, RSR, ROR) that closely correspond to their research questions. The experimental analysis based on these metrics reveals several interesting conclusions, for example:
    - For some models, the impact of the NDR metric is quite significant. Furthermore, some model families (like Llama) exhibit a trend where larger models show worse NDR robustness compared to smaller ones. I think NDR can be a robustness metric for evaluating base model.
    - The experiment finds that for most models the reversed document order yields better results. This aligns with findings on recency bias in prompt engineering and is a practical discovery.
    - The experiment in Section 5.5 on different prompting strategies is also interesting, although I have a different interpretation of the results than the authors (see "Questions").

**Weaknesses:**

1. I think there is a lack of rigor in contribution positioning and over-claiming in this paper:
    - The authors' core argument is that "previous benchmark work used a large amount of artificially synthesized/constructed noise data, and is therefore not realistic". However, this argument ignores that the "real internet" retrieval environment is itself flooded with a large amount of noise, errors, and misinformation. From this perspective, previous work (e.g., inserting counterfactual noise) can be seen as a method of simulating this "real internet noise" in a controlled laboratory environment.
Conversely, the "pure Wikipedia" retrieval corpus constructed by this paper—a highly fact-checked and relatively "clean" corpus—is, in fact, a less realistic scenario for simulating open internet retrieval. A model retrieving from the real internet cannot possibly consider only clean Wikipedia text while completely ignoring other noisy websites.
    - I think in work pre-dating 2023 (i.e., before the rise of strong LLMs), evaluation under a "natural retrieval top-k" setting would not be novel. Does this paper clearly articulate its true position within the developmental timeline of RAG evaluation? A more accurate contribution of this paper might be positioned as: "Re-visiting the 'natural retrieval' setting in the era of strong LLMs, using new metrics (like NDR) to evaluate the resulting knowledge conflict problem".

2. The paper's evaluation focuses primarily on "non-thinking" models. However, current SOTA LLMs' robustness have been widely boosted with their thinking abilities. I think including more thinking-based SOTA LLMs in the experiments and analysis would benefit the paper and improve the validness.

**Questions:**

1. In Section 5.5, the authors conclude that "using OwnKnow might limit the maximum performance models can possibly achieve". I have a different opinion: observing the results, for smaller models, OwnKnow generally shows a decrease in performance, but for larger and better models, OwnKnow often leads to a performance improvement. I find this point to be intuitive and reasonable. As the model's capability enhances, it can rely more on its own knowledge, or in other words, when the model has stronger discernment ability, referencing its own knowledge should always be beneficial. This is analogous to how humans with a certain degree of cognition and discernment understand the world / answer questions based on reference materials.

2. If the author can better declare their contribution and well address my concerns in Weakness point 1 with rigorous justification, I will adjust my assessment.

---

> ### Author Response · Authors · 2025-12-04
>
> > Re Weakness 1: The authors' core argument is that "previous benchmark work used a large amount of artificially synthesized/constructed noise data, and is therefore not realistic". However, this argument ignores that the "real internet" retrieval environment is itself flooded with a large amount of noise, errors, and misinformation. From this perspective, previous work (e.g., inserting counterfactual noise) can be seen as a method of simulating this "real internet noise" in a controlled laboratory environment. Conversely, the "pure Wikipedia" retrieval corpus constructed by this paper—a highly fact-checked and relatively "clean" corpus—is, in fact, a less realistic scenario for simulating open internet retrieval. A model retrieving from the real internet cannot possibly consider only clean Wikipedia text while completely ignoring other noisy websites.
> > I think in work pre-dating 2023 (i.e., before the rise of strong LLMs), evaluation under a "natural retrieval top-k" setting would not be novel. Does this paper clearly articulate its true position within the developmental timeline of RAG evaluation? A more accurate contribution of this paper might be positioned as: "Re-visiting the 'natural retrieval' setting in the era of strong LLMs, using new metrics (like NDR) to evaluate the resulting knowledge conflict problem".
>
> We appreciate the detailed comments. We believe the definition of a "realistic" setup depends largely on the target application.
> The demand for RAG robustness is strongest in applications requiring high reliability (e.g., finance and healthcare). In these scenarios, systems are typically restricted to retrieving from credible, curated sources rather than the open internet.
> Consequently, for the users most concerned with robustness, a "clean" corpus containing natural contradictions is a far more realistic proxy than a dataset simulating open-web noise or artificial errors.
> We agree that our contribution is best framed as evaluating these dynamics in the modern LLM era. In the revision, we will position our paper as: "Re-visiting RAG robustness with trusted sources in the era of strong LLMs" to better contextualize our focus.
>
> > Re Weakness: 2 The paper's evaluation focuses primarily on "non-thinking" models. However, current SOTA LLMs' robustness have been widely boosted with their thinking abilities. I think including more thinking-based SOTA LLMs in the experiments and analysis would benefit the paper and improve the validness.
>
> Thanks for the suggestion. Our experiments have 108 setups for each model and each setup requires inference over all 1500 samples in the whole benchmark. Due to the budget limit, we were only able to experiment with o3-mini (a thinking model). We will consider adding more thinking models, e.g., qwen, in the revision.
>
> > Re Question: In Section 5.5, the authors conclude that "using OwnKnow might limit the maximum performance models can possibly achieve". I have a different opinion: observing the results, for smaller models, OwnKnow generally shows a decrease in performance, but for larger and better models, OwnKnow often leads to a performance improvement. I find this point to be intuitive and reasonable. As the model's capability enhances, it can rely more on its own knowledge, or in other words, when the model has stronger discernment ability, referencing its own knowledge should always be beneficial. This is analogous to how humans with a certain degree of cognition and discernment understand the world / answer questions based on reference materials.
>
> Our interpretation focuses a lot on the most robust model (o3-mini) and we observe a decrease in the best performance across our setups.
> We think that your interpretation also sounds reasonable and there could be many open discussions. For example, o3-mini seems to be a counter example to your interpretation, but given that it is a reasoning model, could it be that the reasoning capability might it less adhering to its own knowledge?

---

### Official Review · Reviewer_9zah · 2025-11-01

**Soundness:** 2
**Presentation:** 2
**Contribution:** 2
**Rating:** 2
**Confidence:** 4

**Summary:**

This paper studies how reliably large language models perform in retrieval-augmented generation (RAG) under realistic settings. The authors build a benchmark of 1,500 open-domain QA samples with Wikipedia retrievals and define three metrics: No Degradation Rate, Retrieval Size Robustness, and Retrieval Order Robustness. The authors aim to measure whether RAG consistently helps, scales with more documents, and remains stable across document orders. Testing 11 LLMs with different prompting strategies, they find that most models are robust in practice but still show inconsistencies that limit RAG’s full benefits. The work offers a practical benchmark, clear robustness metrics, and comprehensive insights for improving dependable RAG systems

**Strengths:**

1) The paper defines three clear quantitative metrics—No Degradation Rate, Retrieval Size Robustness, and Retrieval Order Robustness—that formalize intuitive aspects of retrieval robustness in RAG systems.

2) It provides a practical benchmark of 1,500 open-domain QA samples using Wikipedia retrievals and two standard retrievers (BM25 and BGE), offering a reproducible setup grounded in real-world retrieval conditions.

3) The experimental coverage is broad, evaluating 11 LLMs and three prompting strategies, giving a comprehensive empirical picture of current RAG robustness across models.

**Weaknesses:**

1) The novelty is limited. The core idea is primarily a large-scale empirical study rather than a methodological or theoretical innovation. The proposed metrics formalize well-known intuitions but do not introduce new techniques or insights into why robustness varies.

2) The benchmark focuses only on open-domain QA with Wikipedia, which restricts generalization to specialized domains or other RAG use cases such as reasoning, dialogue, or summarization.

3) The evaluation relies heavily on one judging model (Llama-3.3-70B-Instruct), yet potential biases or inconsistencies across evaluators are not analyzed. No statistical significance tests are reported

4) The analysis of retrieval size and order is shallow. It tests only three orderings (original, reversed, shuffled) and does not explore learned or adaptive re-ranking, nor deeper interactions between retrieval noise, context length, and answer quality.

5) The reproducibility and release timeline are unclear since code and scripts are promised only after acceptance, preventing full verification during review.

**Questions:**

- How consistent are robustness scores when using different evaluator models? Have you checked for systematic bias toward any LLM family?

- Why were only three simple document orderings tested? Would a learned or heuristic re-ranking strategy affect the results?

- Can you provide finer-grained analyses of which samples lose performance under larger retrieval sizes or different orders?

- What are the trade-offs of the OwnKnow and S2A prompting strategies across models and datasets?

- How sensitive are the proposed metrics to evaluation noise or variation in the scoring model? Can you report confidence intervals or variance across repeated runs?

---

> ### Author Response · Authors · 2025-12-04
>
> > Re Weakness 1: The novelty is limited. The core idea is primarily a large-scale empirical study rather than a methodological or theoretical innovation. The proposed metrics formalize well-known intuitions but do not introduce new techniques or insights into why robustness varies.
>
> The study of our paper shows that robustness is less of a concern for RAG systems with credible sources like Wikipedia, while previous studies on RAG robustness focuses on artificial cases with adversarially edited sources. Additionally, we believe it is important to formalize intuitions.
> We would also like to highlight other reviewers' acknowledgment of our novelty, including "The research questions are well-chosen and address core concerns of RAG practitioners." from Reviewer xwqz and "this paper’s setup (strong LLM + natural top-K retrieval) helps us, in the current stage of LLM development, to re-evaluate the robustness of SOTA LLMs in retrieval-based QA" from reviewer gXTC.
>
> > Re Weakness 2: The benchmark focuses only on open-domain QA with Wikipedia, which restricts generalization to specialized domains or other RAG use cases such as reasoning, dialogue, or summarization.
>
> Please see our general response.
>
> > Re Weakness 3, Question 1, and Question 5: The evaluation relies heavily on one judging model (Llama-3.3-70B-Instruct), yet potential biases or inconsistencies across evaluators are not analyzed. No statistical significance tests are reported.
>
> Please see our general response, where we show results evaluating with different judges and with the same judge with different random seeds.
> Due to the large-scale experimental setup, applying this variance study to the final robustness metrics would be very expensive, but we think the low variance of the judge outputs can mitigate the concern over metric variance.
>
> > Re Weakness 4 and Question 2: The analysis of retrieval size and order is shallow. It tests only three orderings (original, reversed, shuffled) and does not explore learned or adaptive re-ranking, nor deeper interactions between retrieval noise, context length, and answer quality.
>
> While we agree that adding more orderings would be helpful, we would like to point out that the scale of experiments increases a lot every time we add a new choice in one of the studied factors, which can become prohibitive due to the budget limit.
> Specifically, adding a new order will incur 2 retriever $\times$ 6 retriever size $\times$ 3 prompting strategies $\times$ 11 models = 396 model runs over the whole benchmark (1500 questions).
>
> And "deeper interactions between retrieval noise, context length, and answer quality" usually require manually design the evaluation setups which some previous studies did. In this study, we would like the setup to be realistic and the retrieval noise comes up naturally from commonly used retrieval setups.
>
> > Re Weakness 5: The reproducibility and release timeline are unclear since code and scripts are promised only after acceptance, preventing full verification during review.
>
> Thanks for pointing out this issue. We are actively pushing for getting the evaluation harness released. Redistribution Wikipedia is a legal bottleneck and we have to revamp some setups to make it compatible with an open-sourced and licensed Wikipedia corpus. We will deliver it as soon as we can.
>
> > Re Question 3: Can you provide finer-grained analyses of which samples lose performance under larger retrieval sizes or different orders?
>
> Thanks for the suggestion. We will provide this analysis in the next version.
>
> > Re Question 4: What are the trade-offs of the OwnKnow and S2A prompting strategies across models and datasets?
>
> We discussed the trade-off in Line 415--424 and Line 429--431.
> We don't see a consistent improvement in robustness when using the S2A prompting strategies.
> The OwnKnow strategy consistently improves the robustness, while it could limit the maximum performance the model can achieve across various setups.

---

### Official Review · Reviewer_xwqz · 2025-11-02

**Soundness:** 3
**Presentation:** 4
**Contribution:** 3
**Rating:** 6
**Confidence:** 4

**Summary:**

The paper proposes a sample-level metric to measure retrieval robustness and conducts insightful analysis. The paper is easy to follow. However, the practical utility is limited by a relatively narrow experimental setting focused on knowledge-intensive QA with Wikipedia knowledge base. Another question is can the proposed metrics be enhanced to better capture the observed performance trade-offs across individual samples?

**Strengths:**

1. The paper proposes a sample-level metric to measure retrieval robustness, which is highly relevant for the practical deployment of RAG systems. The research questions are well-chosen and address core concerns of RAG practitioners. Comprehensive experiments are conducted with 11 modern LLMs with 3 prompting strategies. The work gives insight of how modern LLMs react to external information with various quality.

2. Detailed experiment design! I really appreciate the analysis shown in Sec.5.3 "models keep trading off performance across individual samples...imperfect robustness on retrieval size.". (Also Sec.5.4)

3. Paper writing is easy to follow.

**Weaknesses:**

1. The experiment scopes remain limited on 1) knowledge-intensive open-domain QA tasks. 2) limited knowledge base for retrieval. The two limits limit the practical benefit of the research. Robustness on more domains and more various settings are encouraged. E.g., how the three metrics will be on Google Search? How the LLMs will behave if the queries are a blend of knowledge-intensive ones and non-knowledge-intensive ones. The good robustness of models on knowledge-intensive queries with Wikipedia as knowledge base is natural.

**Questions:**

1. As stated in Sec.5.3 and 5.4, cases where "...hurting performance on some examples while gaining performance on others." are observed. Is there a way to enhance the three metrics NDR, RSR and ROR, to include this phenomenon?

---

> ### Author Response · Authors · 2025-12-04
>
> > Re Weakness:  The experiment scopes remain limited on 1) knowledge-intensive open-domain tasks. 2) limited knowledge base for retrieval.
>
> Please see our general response.
>
> > Re Question: As stated in Sec.5.3 and 5.4, cases where "...hurting performance on some examples while gaining performance on others." are observed. Is there a way to enhance the three metrics NDR, RSR and ROR, to include this phenomenon?
>
> Our metrics already penalize this phenomenon, as we measure robustness (whether the performance drop) at the sample level instead of at the dataset level.

---

### Author Response · Authors · 2025-12-04
**General Response**

We thank the reviewers for their feedback!

We appreciate that all reviewers (xwqz, 9zah, gXTC, VDE5) commends our comprehensive experimental design, describing it as a "thorough and insightful empirical study" that offers a "comprehensive empirical picture" of the current RAG landscape.
They also unanimously praised our proposed metrics (NDR, RSR, ROR), noting they "formalize intuitive aspects of retrieval robustness" via a "rigorous and standardized methodology."
Additionally, Reviewers 9zah, gXTC, and VDE5 valued our benchmark's practical utility, highlighting its "reproducible setup grounded in real-world retrieval conditions" versus artificial noise.
Finally, Reviewers xwqz, gXTC, and VDE5 appreciated our "insightful analysis" regarding model performance trade-offs.

**Regarding reviewer 9zah and VDE5's concern over using a single LLM as the judge model**

First, we note that the answers we are evaluating are short-form and previous work mainly uses exact match to evaluate the correctness of the answers.
Exact match is flawed, so we choose to use an LLM as the judge to evaluate the correctness of the answers based on semantic similarity,
and llama-3.3-70b is strong enough to check semantic similarity between ground-truth and model-generated answers.

That being said, we have added experiments evaluating outputs generated by GPT-4o, Llama-3, and Claude-3.5 using GPT-4o and Llama-3 as judges. We compare the evaluation by the two judges on Natural Questions and Hotpot QA.

As shown in the table below, while Llama-3 tends to predict more positive labels, the two judges agree well on the performance gap between models, and we thus think there is no bias towards a certain model family that would impact the trend in our results.

Natural Questions

| RAG Model \ Judge Model | GPT-4o | Llama-3 |
| --- | --- | --- |
| GPT-4o | 74.60 | 80.60 |
| Llama-3 | 64.60 | 71.00 |
| Claude-3.5 | 68.40 | 76.20 |

Hotpot QA

| RAG Model \ Judge Model | GPT-4o | Llama-3 |
| --- | --- | --- |
| GPT-4o | 57.80 | 65.60 |
| Llama-3 | 44.00 | 50.40 |
| Claude-3.5 | 56.20 | 62.00 |

We have also added experiments that run the Llama-3 judges with 5 different random seeds on the outputs generated by GPT-4o, Llama-3, and Claude-3.5. We report the 95\% confidence interval in the table below, showing that the judge is very robust.

| | Natural Questions | Hotpot QA |
| --- | --- | --- |
| GPT-4o | $81.08 \pm 0.33$ | $65.52 \pm 0.22$ |
| Llama-3 | $70.64 \pm 0.48$ | $50.12 \pm 0.37$ |
| Claude-3.5 | $76.20 \pm 0.39$ | $61.84 \pm 0.21$ |

**Regarding reviewer xwqz, 9zah, and gXTC's concern over only using Wikipedia as the retrieval corpus and only involving knowledge-intense questions**

We acknowledge that extending the study to more diverse retrieval corpora would be helpful and we will discussion this limitation in the revision.
Nonetheless, we think the major use case of RAG is answering knowledge-intense queries.
We choose Wikipedia as the retrieval corpus to ensure reproducibility and trustworthiness of the retrieval corpus, which aligns with robustness-focused applications that would retrieve only from credible sources rather than any source.

---

### Meta-Review · Area_Chair_uiCn · 2026-01-06

**Summary:**

The major remaining concerns include the following:
* the novelty of the benchmark paper: this paper does not carefully compare existing RAG benchmark datasets as well as RAG methods to tackle the similar robustness issues.
* the slightly limited experiment scope: The paper experiment setup is quite extensive but most reviewers agree to expand the setup toward more interesting directions (e.g., diversity in models and robust judge models)

**Reviewer Concerns:**

**Reviewer xwqz**:
The reviewer raised two concerns, which are partially addressed as follows:
1. (*limited experiment scope*) The experiment scopes remain limited on 1) knowledge-intensive open-domain QA tasks and 2) limited knowledge base for retrieval – limitations are acknowledged.
2. (clarification on metrics) Is there a way to enhance the three metrics NDR, RSR and ROR, to include a  phenomenon "...hurting performance on some examples while gaining performance on others."? – only provide a brief answer that does not provide any details.

The limited experiment scope is an outstanding issue, which cannot be addressable during the rebuttal.


**Reviewer xwqz**:
The reviewer introduced 8 concerns and they are only partially addressed.

1. (*limited novelty*) The novelty is limited. The core idea is primarily a large-scale empirical study rather than a methodological or theoretical innovation – partially addressed by pointing other reviewers’ interpretations of novelties and contrasting to previous study, which does not directly mitigate the concerns.

2. (*limited experiment scope*) The benchmark focuses only on open-domain QA with Wikipedia – does not directly address the concern on the limited evaluation over open-domain QA instead of QA from specialized domains.

3. (*biased usage of a judging model*) The evaluation relies heavily on one judging model (Llama-3.3-70B-Instruct), yet potential biases or inconsistencies across evaluators are not analyzed – provide simple experiments that demonstrate that the using a single judging model does not introduce the bias in evaluation without redoing a few experiments with multiple judging models with majority voting.

4. (*no random experiments*) No statistical significance tests are reported – addressed by adding random experiments with 5 trials and showing the variance is low.

5. (*limited analysis*) The analysis of retrieval size and order is shallow – justified by claiming that additional analysis is prohibitive due to the budget limit.

6. (*no code*) The reproducibility and release timeline are unclear – confirmed that code will be released as soon as possible after addressing issues (e.g., Wikipedia license)

7. (*experiments on diverse retrieval sizes or different orders*) Can you provide finer-grained analyses of which samples lose performance under larger retrieval sizes or different orders? –  acknowledged and promised to be added in the next version.

8. What are the trade-offs of the OwnKnow and S2A prompting strategies across models and datasets? – addressed by providing performance differences of OwnKnow and S2A.


The most outstanding and remaining concern is the limited experiment setups, which is usually needed for evaluation papers.


**Reviewer gXTC**:
The reviewer shares two concrete concerns, which are only partially addressed.
1. (*limited novelty*) A lack of rigor in contribution positioning and over-claiming – partially addressed by claiming that Wikipedia is more realistic for some applications (rather than internet documents) but did not provide the developmental timeline of RAG evaluation to contrast the paper’s contributions. .

2. (*limited models*) The paper's evaluation focuses primarily on "non-thinking" models – promised to add additional thinking models during the revision due to the budget limit,

The outstanding and remaining concern is on the novelty of this paper – the authors could not contrast the proposed benchmark paper to existing benchmark papers within the developmental timeline of RAG evaluation.


**Reviewer VDE5**:

The reviewer shared three concerns and they are partially addressed as follows:

1. (*limited novelty*) The paper fails to discuss or compare against "Astute RAG" [1] – not addressed; authors’ should have explained whether the issues that the proposed benchmark is considered in Astute RAG (even though it is a method paper).

2. (*motivation in building a new dataset*) The rationale for curating a new, small-scale benchmark (1500 samples) from existing datasets is unclear – addressed by explaining that this paper uses subsample due to the computational time with a larger number of setups.

3. (*limitation in evaluation*) The reliability of the results is questionable, as they depend on Llama3.3-70B-Instruct for correctness judgments – partially addressed by showing a simple experiment on the unbiasedness of judge models.

The most outstanding and remaining concern is the novelty of the proposed benchmark – for example, Astute RAG attacks a similar issue (i.e., imperfect retrieval robustness); even though it is not a benchmark paper, the missing reference and discussion suggests that the paper lacks exploration on tightly related work.

**Reviewer Scores:**

**Reviewer nenT**:
Final expected rating: 6 / final expected confidence: 4 – The raised concerns are only partially addressed, so I guess the reviewer will maintain scores.

**Reviewer xwqz**:
Final expected rating: 2 / final expected confidence: 4 – The most concerns may not be addressable during the rebuttals; thus, I expect the reviewer would maintain its score.

**Reviewer gXTC**:
Final expected rating: 4 / final expected confidence: 3 – The concerns are only partially addressed, so the reviewer might maintain its score.


**Reviewer VDE5**:
Final expected rating: 2 / final expected confidence: 3 – The concerns are only partially addressed, so the reviewer might maintain its score.

---

### Decision · Program_Chairs · 2026-01-26

Reject